# Temperamental Development among Preterm Born Children. An RCT Follow-up Study

**DOI:** 10.3390/children7040036

**Published:** 2020-04-23

**Authors:** Inger Pauline Landsem, Bjørn Helge Handegård, Stein Erik Ulvund

**Affiliations:** 1Child & Adolescent Department, University Hospital of North Norway, 9019 Tromsø, Norway; 2Health Research Faculty, UIT the Arctic University of Norway, 9019 Tromsø, Norway; bjorn.helge.handegard@uit.no; 3Department of Education, University of Oslo, 0317 Oslo, Norway; s.e.ulvund@iped.uio.no

**Keywords:** prematurity, temperament, parenting, RCT design, long-term follow-up

## Abstract

A randomized controlled trial study recruited 146 preterm born children, either to participate in a modified version of the Mother–Infant Transaction Program (MITP-m) or to receive the usual follow-up services, before and after discharge from a neonatal intensive care unit. This follow-up study investigates whether MITP participation is associated with parental perceptions of child temperament from two to seven years. Children’s temperament was reported by mothers and fathers separately at children’s ages of 2, 3, 5, and 7 years. Parents in the MITP-m group reported lower levels of negative emotionality in their children compared to the control group. In maternal reports, a group effect (F(1, 121) = 9.7, *p* = 0.002) revealed a stable difference in children’s negative emotionality from two to seven years, while a group-by-time interaction related to an increasing difference was detected in reports from fathers (F(1, 94) = 4.8, *p* = 0.03). Another group difference appeared in fathers’ reports of children’s soothability (F(1, 100) = 14.2, *p* < 0.0005). MITP-m fathers seemed to perceive their children as easier to soothe at all ages as no interaction with time appeared. Parental reports on children’s sociality, shyness, and activity did not differ between the groups.

## 1. Introduction

Severe prematurity is associated with long-term developmental difficulties, and other factors such as heredity, health problems, and environmental conditions influence the extent of problems experienced [1,2]. Previous studies have documented that temperamental difficulties reported in preterm born children (hereafter named preterms) affect later outcomes such as children’s self-regulatory capacities [3,4,5], behavior problems, social skills [4,6,7], and language development [8,9]. In addition, a difficult temperament is associated with more parenting stress [10]. Overall, it seems evident that high reactivity, negative emotionality, weak effortful control, and low self-regulation capacity are risk factors that put preterms at increasing risk across childhood [4,10,11,12,13].

Most conceptualizations of temperament agree that it is biologically based and has relatively stable features, even though influenced by children’s maturation, learning, and environmental transactions [14,15]. A child’s temperament covers the unique way emotions are expressed, characterized by latency, intensity, and duration of responses [16]. The number of temperamental constructs and the terms used vary, depending on definitions and measures [17]. 

There have been calls for studies that investigate how challenging temperamental expressions among preterms may be met by environmental adjustments [4,13,18]. Some studies have focused on neonatal factors (e.g., less pain exposure) [18,19] and the risk of altered brain maturation, since preterms may be exposed to non-nutritional stimuli in a hospital unit for months [20]. Others point out that sensitive and supportive parenting is particularly important for children with challenging temperament dispositions [3,21,22] as it is for preterms [23,24,25]. Lastly, it is reported that factors such as the child’s age and gender and maternal factors (e.g., age and education) may influence temperamental development [26,27].

According to the transactional model of development, child and parent characteristics and their dyadic and triadic interactions develop over time due to mutual influences [28]. Several studies have focused on preterms’ susceptibility to parenting [11,29]. Furthermore, parenting is described as more challenging after a preterm birth, because of the abruptness of the preterm delivery [30], the parent–child separation [31], unfamiliar early surroundings [32], and these children’s often poorly regulated behavior that may seem incomprehensible to new parents [22,33]. In families with very preterm children, parent–child dyads have previously been described as having less frequent symmetric co-regulation patterns and less positive and more neutral affective intensity of both infants and mothers, compared to families with full-terms [34]. Thus, prematurity, children’s temperament, and parents’ capacity to make suitable adaptations to their children represent a mix of factors that may influence long-term transactional patterns. Early interventional effects on parenting stress have been reported [35], and parenting stress is assumed to affect parents’ capacity to be sensitive [36,37]. Mothers of preterms are reported to be equally sensitive as mothers of full-terms [38], but preterms seem more negatively affected by low parental sensitivity than full-terms [24,37,39,40]. Thus, parenting stress reported in infancy may be a predictor of parents’ perception of preterm born children’s temperament [21]. 

This paper describes part of a randomized clinical trial (RCT) that aimed to determine whether an early, structured parental guidance program could improve developmental outcomes in a group of very preterm born children [41,42]. Temperament was reported at similar levels in the intervention and control groups in infancy [42]. One early finding was that the program positively influenced the association between child regulatory behavior and maternal stress reported at six months and one year [42]. The current focus is on how parents’ perception of children’s temperament has developed across toddler and preschool years. This study explores whether dimensions of emotionality, activity, sociability, shyness, and soothability differ between an intervention and a control group from two to seven years, and to what extent differences may be associated with parental participation in an early parent training program. It is hypothesized that the modified Mother–Infant Transaction Program (MITP-m) empowered parents, in addition to decreasing parenting stress reported by mothers at all ages. Mothers in the MITP-m group reported a higher perception of competence in their parenting role at children’s ages of one, two, three, five, and seven [35]. This may have enhanced their ability to adapt to their child’s temperament more successfully than the control parents. 

The following question will be addressed: Is the development of temperamental constructs (emotionality, activity, sociability, shyness, and soothability), as reported by parents of preterms, positively associated with participation in an early parent training program?

## 2. Methods

### 2.1. Participants

Preterms (birthweight <2000 g), born between March 1999 and September 2002 were randomized, either to participate in a structured parent training program (PI group, *n* = 72) or to follow-up as usual (PC group, *n* = 74). Written parental consent was collected twice, before discharge from the neonatal intensive care unit (NICU) and before the seven-year follow-up session. Children without congenital anomalies and whose mother spoke Norwegian were eligible. In addition, a group of healthy full-term children was recruited. Data from these families are not included in this study. Power calculations were based on a focus on cognitive outcomes at age two [41], but retrospective power calculations for this RCT follow-up study were not performed. A complete diagram showing the flow of participants through each stage of the study has previously been published [35,41,43] and is available here as additional online information. Families in both groups received services in line with the NICU guidelines for discharge and follow-up of patients. These included medical examinations, a session with a child physiotherapist, information about nutritional supplements, and community follow-up services. Later follow-up services from the hospital varied depending on the child’s degree of prematurity and perinatal complications. Randomization ensured well-balanced groups with one exception, namely that mothers in the PI group had on average 1.1 more years of education (Table 1). The groups did not differ in relation to parents’ marital status, income, or ethnicity.

During the study, all children and their parents participated in the assessment program with various tests and questionnaires at corrected ages of 6 months and 1, 2, 3, 5, and 7 years. Withdrawal rates were low: 89% of the preterms were still participating at seven years (Appendix A, additional resources). The study has repeatedly been approved by the Regional Committee for Medical and Health Research Ethics (REK Nord) and the Norwegian Data Protection Authority (1999, 2005, and 2010) and the ClinicalTrials.gov registration is NCT00222456. All subjects gave their informed consent for inclusion before participation. The study was conducted in accordance with the Declaration of Helsinki, and the protocol was approved by REK Nord (2018/1966). As the children involved in this study now are more than 16 years old, they were also given written information about this study and a possibility to withdraw from it (in line with ethical research policy in Norway). One person wanted to be excluded from this study sample and was therefore deleted from data used in this analysis.

### 2.2. Intervention Program

The parent training program is a modified version of the Mother–Infant Transaction Program (MITP-m). It started in the last week of the hospital stay and ended three months after discharge [44]. The MITP-m diverged from the original MITP in two ways: (a) inclusion of an initial alliance-building session, where parents were invited to express their experiences from the pregnancy, delivery, and hospital stay, and (b) no written summary of the MITP-m sessions was given to parents when the program ended [44]. The MITP-m consisted of eight one-hour sessions in hospital, focusing on the individual child’s autonomic, motor, state and social functioning and his/her regulatory capacities. Behavioral characteristics (e.g., physiological signs of stress, reflexes, state transitions, social availability) of each child were studied through shared observations and caring actions with his/her parents. The sessions aimed to empower parents and help them to realize how they could adjust caring activities to support their child’s overall regulatory capacities. After discharge, each PI family received four home visits after one, two, four, and twelve weeks. In these sessions, all topics related to the children’s regulation of autonomic, motor, state, and social functioning were discussed in detail, and changes (e.g., less need of breaks, less sensitive to sounds and light) in need of parental co-regulation and support were discussed. Parents’ learning and successful adaptation to their children’s maturation were praised. In the third home visit, parents’ perception of the child’s early temperamental expressions, such as activity, rhythmicity, intensity, thresholds, mood, and sociality, were addressed [44]. This was followed by shared reflections about how these behaviors could be interpreted. Mothers participated in all MITP-m sessions. Fathers participated in six of twelve sessions on average, and 65% of them participated in three or four of the home visits. All interventionists were specially trained nurses, each family was guided by the same nurse in all sessions, and all sessions were described in logs reviewed by the study director after completion. 

### 2.3. Measures

Children’s temperament was reported by mothers and fathers separately at children’s ages of 2, 3, 5, and 7 years. The questionnaire used in the current study consists of seven dimensions, either previously described in the Emotionality, Activity, and Sociability (EAS) questionnaire [45] or in the Colorado Childhood Temperament Inventory (CCTI) [46]. Questions tapping children’s emotionality concern whether the child tends to cry easily, often fusses and cries, gets upset easily, tends to be emotional, and reacts intensely when upset. The soothability dimension covers descriptions such as “whenever the child starts crying, he can be easily distracted”, “if talked to, child stops crying”, and “child tolerates frustration well”. The questionnaire was translated and adapted for this study by a Norwegian psychologist. The questionnaire consists of 35 questions, 24 of which relate to the five dimensions reported. Five items were loaded on each of the Emotionality, Activity, Sociability, and Soothability dimensions, while four items could be included in the Shyness dimension. All items were assessed on a five-point scale ranging from “not typical” to “very typical”. An analysis of reliability showed that a fifth Shyness item had changed its meaning during translation, and it was therefore excluded. Reliability tests for each dimension at different ages were calculated (Table 2). Most Cronbach’s alphas varied between 0.67 and 0.72, but in paternal reports of Sociability, the alphas fell below 0.60 at the ages of 2, 3, and 5 years (Table 2). A high proportion of participating parents responded to the questionnaires: at 2 years (PC/PI mothers: *n* = 67/59; PC/PI fathers: *n* = 59/49), at 3 years (PC/PI mothers: *n* = 67/65; PC/PI fathers: *n* = 61/53), at 5 years (PC/PI mothers: *n* = 68/60; PC/PI fathers: *n* = 59/52), and at 7 years (PC/PI mothers: *n* = 67/64; PC/PI fathers: *n* = 55/53). 

Information about children’s birth status, medical treatment, and parent-reported socio-demographic conditions was collected before discharge from hospital. Parental stress was monitored across all follow-ups using the Parenting Stress Index (PSI), third edition [36]. The PSI consists of 123 items that cover three main dimensions: child- and parent-related stress and stress related to challenging life events. In this report, two variables are included. First, a parenting stress variable was included, based on both PSI parent- and child-related stress (101 items) reported at children’s age of one. Secondly, we included a parental education variable, defined as years of education at inclusion in the study. 

### 2.4. Data Analysis

Differences in temperament related to the study groups and time were studied using IBM SPSS statistics (version 25) and linear-mixed model analysis, with full-information maximum-likelihood estimation. All data were thus included in the analysis. Separate analyses were conducted for each dimension of temperament and for reports from mothers and fathers, thus ten analyses were performed. Time was treated as a continuous variable, coded as the number of years from baseline, and baseline was defined as the measure of temperament at age two (time coded as 0, 1, 3, 5). Intercepts were allowed to vary between subjects, and a random slope for time was included in the model. The level of significance was 0.05. 

## 3. Results

### 3.1. Child Temperament Reported by Mothers

No group-by-time interactions were detected in reports from mothers (Table 3). A group difference appeared in mothers’ reports on children’s emotionality. PC mothers reported child emotionality as significantly higher than PI mothers did at age two, corrected for maternal stress reported at age one and maternal years of education (Table 3). This difference persisted until children’s age of seven as no interaction with time appeared. No significant group differences appeared in mothers’ ratings of sociability, shyness, or activity. Group means of sociability and shyness were stable across time, while mothers reported decreasing levels of activity from toddlerhood to early school age in both groups (time effect: F(1, 109) = 70.5, *p* < 0.0005). Appendix A (additional resources) shows how covariates (maternal stress at age one and years of maternal education) affected the analysis. 

### 3.2. Child Temperament Reported by Fathers

There was a significant group-by-time interaction for children’s emotionality in toddlerhood (F(1, 94) = 4.8, *p* = 0.03). The PI and PC fathers reported similar levels of children’s emotionality in toddlerhood with increasing differences over time. PI fathers reported decreasing levels of emotionality with increasing age, while PC fathers reported emotionality at more stable levels from toddlerhood to early school age (Table 3, Figure 1). Children’s soothability was reported as higher by PI fathers than by PC fathers at age two F(1, 100) = 14.2, *p* < 0.0005, and this difference continued with increasing age (Table 3). Lastly, both PC and PI fathers reported their children’s shyness and sociability as stable across childhood, while activity was reported with decreasing scores in both groups (F(1, 91) = 61.7, *p* < 0.0005). Appendix A (additional resources) shows how covariates (paternal stress at age one and years of paternal education) affected the analysis. 

## 4. Discussion

This study explores parental perception of temperamental development among preterms across childhood. It investigates to what degree parents’ participation in an early, structured parent training program influences their perception of temperamental patterns from two to seven years. The main finding is that preterms’ emotionality and soothability were positively associated with participation in the MITP-m, while preterms’ shyness, sociability, and activity showed similar development in both groups. Mothers’ and fathers’ perceptions of preterms’ shyness and sociability were reported with high mean group stability across early childhood, while children’s levels of activity decreased from toddlerhood until early school age. On the other hand, as explained below, mean levels of children’s emotionality and soothability as reported by fathers formed somewhat different trajectories, depending on the group of the families. This created some group differences that persisted until age seven. 

Emotionality, as defined by the EAS questionnaire, consists of questions that capture children’s tendency to react with strong negative emotions [15,47]. Thus, while some degree of emotionality will characterize most children, higher scores may be difficult for parents to handle [24,48]. PI mothers reported less emotionality in their children than did PC mothers at the age of two, and this difference persisted at later follow-ups. On the other hand, PI and PC fathers reported emotionality at similar levels in toddlerhood, but while PI fathers reported less emotionality with age, PC fathers continued to rate their children’s emotionality as high until age seven (Figure 1). 

In the MITP-m program, the concept of temperament was particularly addressed in the third home visit. This was approximately one month after discharge from the hospital, at a time where parents had a more nuanced perception of their child’s unique behavior. Even though many PI fathers could not join the sessions before discharge, due to long distances from home to hospital, older siblings, or work, they were more frequently present for the home visits. PI fathers may have had a catch-up effect from the four home visits, because each visit included reflections about the child’s signs and needs, which may have changed since the previous meeting. Although PI fathers rated their children’s emotionality at similar levels as PC fathers in infancy, they may have acquired knowledge and skills that helped them to cope more successfully in the long term. Thus, the intervention may have strengthened PI parents’ ability to adapt to their children’s emotional expressions and needs early in childhood. This assumption corresponds with previous results from this study. Olafsen et al. found a strong negative correlation between maternal stress and children’s regulatory competence reported by PI mothers at six months, while PC mothers reported a similar association at age one [42]. The authors presumed that the intervention had changed the relationship between maternal stress and children’s temperament already in infancy. The current study indicates that this alteration of transactional patterns may have continued across the following early childhood years. It may also have increasingly influenced fathers’ perception of child temperament, as fathers became more involved in the care of their child after the first year. The Norwegian parental/maternity leave at that time was mostly used by mothers. 

The findings also concur with another study that focused on how an early intervention was especially positive for preterms with negative emotionality in infancy [23]. Those children showed a clinically meaningful decrease in social and cognitive problems three years after an intervention, compared to children with less negative emotionality [23]. Enabling parents to manage their infants’ negative emotionality therefore seems to be a core element in early interventions. Negative emotionality challenges parents’ coping abilities, as their efforts to interact with the child frequently do not seem to work. Many spontaneous parent−child interactions may fail, and some parents may need more knowledge and coping strategies to understand why their child behaves in a particular way. A significant association between fathers’ report of emotionality and years of paternal education indicates that this may have been particularly challenging for fathers with fewer years of education.

How negative emotionality could best be approached was addressed in a recent experimental study [49]. Mothers were asked to soothe a distressed infant simulator in early infancy, and this was compared with the mothers’ report of negative affectivity in their own child in later infancy. Specific behaviors, such as greater use of soothing touch and maternal vocalization in the simulation test were associated with less negative emotionality and fear in the mothers’ children [49]. The importance of touch and vocalization were repeatedly mentioned and praised during the entire MITP-m program. The association discussed here confirms the need for a focus on temperament in interventional research and implementation [23]. 

PI fathers rated their children with higher soothability than PC fathers did at age two, and this difference persisted across follow-ups until children’s age of seven. Fathers’ perception of their children’s soothability depends to a large extent on paternal experiences, knowledge about different strategies, and how each strategy may fit their child’s needs in different situations. Father−child interactions have also been described as more physical, stimulatory, and playful than mother−child interactions [50,51]. In the current study, PI fathers received guidance in the MITP-m home visits about their child’s limited capacity to engage in intense activities. This enabled them to adopt milder ways to soothe their child and adapt this to new interactions as the child grew older. PC fathers, on the other hand, were limited to the strategies they naturally had available based on cultural traditions and their own experiences. This may have been challenging, especially for those caring for a prematurely born child with limited regulatory capacities. PC fathers may to a greater extent have perceived their child as being/having a problem, compared to PI fathers, who may have been empowered by the MITP-m intervention. This suggestion is supported by a previous study which showed more child-related stress reported by PC fathers than by PI fathers until children’s age of five, while paternal stress related to the parenting role was reported at similar levels by all fathers at all ages [35]. By contrast, PC mothers reported more parenting stress than PI mothers at all follow-ups [35], and maternal stress at children’s age of one was a significant covariate in the analysis of maternal perception of children’s soothability (Appendix A, additional resources). 

A child’s availability for parental co-regulation and support is to some extent covered in the soothability dimension in the questionnaire used, identical with the framing in the previously mentioned CCTI questionnaire [46]. Children’s soothability has been reported to moderate associations between negative child temperament and maternal sensitivity [52], between child reactivity and maternal reports of co-parenting quality [53], and described as an important predictor of elevated levels of comorbid internalizing and externalizing behavior problems in referred children [54]. The present findings illustrate that child soothability may be strengthened by a program that probably empowers parents, increases their knowledge, and decreases their sense of stress in parent−child interactions. 

Children’s sociability, shyness, and levels of activity were similar in the two preterm groups, after adjusting for parental level of education and parenting stress in infancy. This study thus confirms the high stability of several temperamental dispositions recently reported in a large sample of American children [26]. In an early work by Buss and Plomin, both emotionality and sociability were described as “superfactors” that seemed to “pervade temperament measures in infancy and early childhood” [45]. Even though these dimensions may be “superfactors”, our study has revealed a potentially important difference between them, namely that preterms’ sociability, shyness, and activity seem unaffected by early environmental changes, while programs such as the MITP-m and others may modify preterms’ emotionality and soothability [23]. This difference is confirmed by the fact that no studies reporting interventional effects on children’s sociability or shyness have been detected despite thorough literature searches. 

Strengths of this study are the RCT design, high participation rates across all follow-ups, and the statistical analysis that allowed all data to be included. Previously validated questionnaires have been used, even though the items originate from two different questionnaires. One limitation is that the original questionnaires were translated without back-translation and one question changed its meaning and was therefore excluded, leaving only four items to cover the shyness dimension. Despite this, a reliability analysis of data reported on each dimension by both mothers and fathers showed moderate internal consistency. In addition, this paper presents results from ten separate analyses, which might imply a possibility of Type I errors in the presentation of results. However, we chose not to perform a Bonferroni correction of the results because it seemed too conservative. To test the robustness of the current findings, this study should be replicated.

## 5. Conclusions

Participation in the MITP-m is associated with parents’ perception of decreased emotionality and, for fathers, increased soothability from two to seven years for very preterm born children. This may be one of the transactional mechanisms behind the long-term positive socio-emotional and behavioral outcomes previously reported from this study [35,43]. Further research in this field should include measures of child temperament. Follow-up programs for very preterm born children should ensure that these elements of the MITP-m are stressed in conversations with parents, and that both mothers and fathers are given equal rights and opportunities to participate in such programs. 

## Figures and Tables

**Figure 1 children-07-00036-f001:**
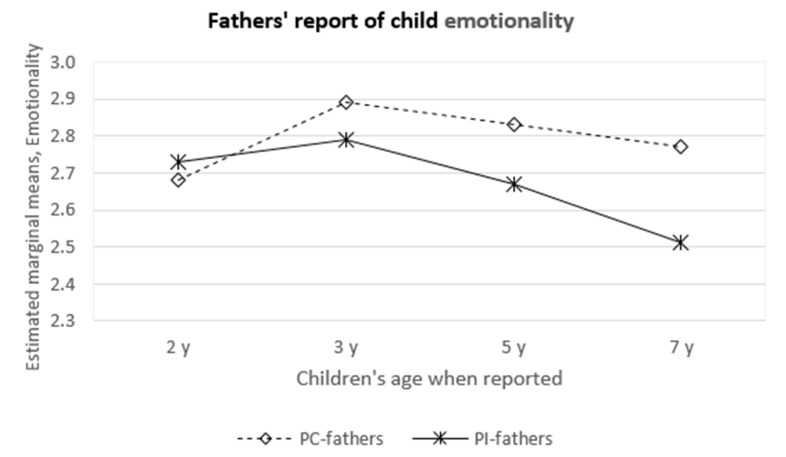
PC and PI fathers’ report of children’s emotionality at different ages.

**Table 1 children-07-00036-t001:** Birth, medical, and demographic information.

	PI Group *n* = 72	PC Group *n* = 74
Infant characteristics		
BW, mean ± SD, g	1396 ± 429	1381 ± 436
400–1000 g, *n* (%)	20 (28)	20 (27)
1001–1500 g, *n* (%)	15 (21)	20 (27)
1501–2000 g, *n* (%)	37 (51)	34 (46)
GA, mean ± SD, weeks	30.2 ± 3.1	29.9 ± 3.5
<28 weeks, *n* (%)	17 (24)	19 (27)
28–32 weeks, *n* (%)	36 (50)	37 (50)
>33 weeks, *n* (%)	19 (26)	18 (24)
Boy, *n* (%)	38 (53)	39 (53)
Twin, *n* (%)	16 (22)	16 (21)
Received ventilation, *n* (%)	29 (40)	37 (50)
Duration of ventilation, *n* (%)	7.0 ± 18.6	7.1 ± 17.3
Postnatal steroid use, *n* (%)	9 (13)	10 (14)
Oxygen therapy at 38 weeks GA, n (%)	11 (15)	14 (19)
Abnormal cerebral ultrasound, *n* (%)		
IVH grade 1 or 2	7 (10)	8 (11)
IVH grade 3 or 4	3 (4)	5 (7)
Periventricular leukomalacia	4 (6)	8 (11)
Maternal and social characteristics		
Mother’s age ^a^, mean ± SD,	30.8 ± 6.1	29.1 ± 6.4
First-born child, *n* (%)	40 (56)	37 (54)
Mother’s education ^a^, mean ± SD, *n* = 131	14.6 ± 2.8	13.5 ± 3.2
Father’s education ^a^, mean ± SD, *n* = 131	13.8 ± 3.1	13.5 ± 3.2
Mother’s monthly income ^b^,	15.8 ± 7.7	14.6 ± 6.7
mean ± SD, *n* = 131		
Father’s monthly income ^b^,	21.1 ± 8.7	19.9 ± 8.1
mean ± SD, *n* = 131		

^a^ years; ^b^ Unit: 1000 Norwegian kroner; BW = birth weight, GA = gestational age, IVH = intraventricular hemorrhage, PI = preterm intervention group, PC = preterm control group, SD = standard deviation.

**Table 2 children-07-00036-t002:** Reliability, dimensions of temperament.

	No. of Items	Maternal Reports Cronbach’s Alpha				Paternal Reports Cronbach’s Alpha			
		T1	T2	T3	T4	T1	T2	T3	T4
**Shyness**	4	0.71	0.65	0.71	0.71	0.67	0.63	0.77	0.69
Emotionality	5	0.67	0.66	0.64	0.66	0.63	0.62	0.69	0.73
Sociality	5	0.62	0.71	0.67	0.76	0.56	0.56	0.57	0.61
Activity	5	0.67	0.67	0.67	0.66	0.66	0.64	0.66	0.63
Soothability	5	0.67	0.72	0.67	0.70	0.71	0.67	0.65	0.69

T1: at children’s age of two years, T2: at three years, T3: at five years, T4: at seven years.

**Table 3 children-07-00036-t003:** Estimated marginal means (and standard errors) and effects related to group and time.

	PI Group				PC Group				Effects Related to Group and Time		
Variables	T1 Mean (SE)	T2 Mean (SE)	T3 Mean (SE)	T4 Mean (SE)	T1 Mean (SE)	T2 Mean (SE)	T3 Mean (SE)	T4 Mean (SE)	Time × Group ^3^	Group ^4^	Time ^5^
Shyness-m ^1^	2.21 (0.09)	2.18 (0.09)	2.23 (0.09)	2.21 (0.09)	2.28 (0.09)	2.20 (0.09)	2.33 (0.09)	2.20 (0.09)	−0.02 ns.	0.02 ns.	<0.01 ns.
Shyness-f ^2^	2.20 (0.08)	2.37 (0.08)	2.28 (0.08)	2.16 (0.09)	2.17 (0.9)	2.20 (0.09)	2.43 (0.09)	2.19 (0.09)	0.02 ns.	0.05 ns.	<0.01 ns.
Emotionality-m ^1^	2.73 (0.08)	2.87 (0.08)	2.67 (0.08)	2.72 (0.08)	2.98 (0.08)	3.02 (0.08)	2.96 (0.08)	2.80 (0.08)	−0.03 ns.	0.33 **	<0.01 ns.
Emotionality-f ^2^	2.73 (0.08)	2.79 (0.08)	2.68 (0.08)	2.50 (0.08)	2.68 (0.08)	2.89 (0.08)	2.83 (0.08)	2.77 (0.08)	0.06 *	0.05 ns.	−0.04 *
Sociality-m ^1^	3.87 (0.08)	3.99 (0.08)	3.84 (0.07)	3.85 (0.08)	3.77 (0.08)	3.95 (0.08)	3.69 (0.08)	3.75 (0.08)	<0.01 ns.	0.08 ns.	−0.02 ns.
Sociality-f ^2^	3.80 (0.07)	3.81 (0.07)	3.70 (0.07)	3.83 (0.07)	3.77 (0.08)	3.66 (0.08)	3.58 (0.08)	3.76 (0.08)	<0.01 ns.	0.07 ns.	<0.01 ns.
Activity-m ^1^	4.07 (0.07)	4.02 (0.07)	3.50 (0.07)	3.60 (0.07)	3.93 (0.07)	3.83 (0.07)	3.44 (0.07)	3.54 (0.07)	0.02 ns.	0.16 ns.	−0.11 ***
Activity-f ^2^	3.97 (0.07)	3.86 (0.07)	3.54 (0.07)	3.43 (0.07)	3.99 (0.07)	3.84 (0.07)	3.70 (0.07)	3.57 (0.07)	0.01 ns.	0.08 ns.	−0.11 ***
Soothability-m ^1^	3.40 (0.07)	3.42 (0.07)	3.39 (0.07)	3.36 (0.07)	3.30 (0.07)	3.13 (0.07)	3.03 (0.07)	3.12 (0.07)	−0.02 ns.	0.17 ns.	−0.01 ns.
Soothability-f ^2^	3.46 (0.07)	3.38 (0.07)	3.36 (0.07)	3.34 (0.07)	3.17 (0.07)	3.02 (0.07)	3.11 (0.07)	3.16 (0.07)	0.03 ns.	0.37 ***	−0.02 ns.

Notes: 1: reported by mothers and 2: reported by fathers at children’s ages of 2 years (T1), 3 years (T2), 5 years (T3), and 7 years (T4). Analyzed with linear mixed models, where 3: estimated difference between slopes for each group; 4: estimated difference between the PC group and the PI group at baseline (2 years); and 5: the effect of time for the PI group. *p* > 0.05 = ns. (not significant), *p* < 0.05 = *, *p* < 0.01 = **, *p* < 0.001 = ***. SE = standard error.

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
