# Peer review of "Temperamental Development among Preterm Born Children. An RCT Follow-Up Study"

_children, 2020, doi:10.3390/children7040036_

Round 1

Reviewer 1 Report

Thank you for the opportunity to review this article. The study presents meaningful questions and results relating to parents' perceptions of children's temperament, their associated parenting stress, and a parenting program designed to improve outcomes for parents and children. I appreciate the inclusion of both mothers and fathers as well as a parenting program with an intentional focus on temperament and parents' understanding of temperament. I would like to see some clarification of concepts as a means of making clear exactly what impact the intervention in purported to have. Primarily, this is a question of whether temperament in this study refers to children's actual behaviors or to parents' perceptions of those behaviors, so that when we read about change (or stability) over time, we understand whether you are suggesting the children are changing or the parents are changing.

i'd like to better understand the relationship between prematurity and what we traditionally think of as temperament. are there certain characteristics that are common to preterm infants that overlap with what we think of as temperament in full term infants? are these characteristics related to prematurity itself (some aspect less developed due to prematurity)? what i'm trying to ask is are we calling something temperament that is somehow different in preterm infants, related to a different biological underpinning than traditional conceptualizations of temperament in full term infants? This may not need to be explained in detail, but it seems like a question a reader might ask. Perhaps, this is just something to think about.

lines 58 through 61, I don't understand how the preceding sentences lead to the conclusion in line 61. Also, what is the distinction between the child's actual temperament and parent's perception of the child's temperament. This seems very important to the argument. Are you arguing that the intervention impacts children's actual behavior (the manifestation of their actual temperament) or that it impacts parents' perceptions of the child's temperament? Clarification of this would be very helpful.

parenting stress - does this mean any sort of stress experienced by parents? or does this refer to Abidin's conception of parenting stress which is in large part a function of parents' perceptions of the difficulty of being a parent and parenting this specific child?

line 65 - the program influenced the association between child regulation and maternal stress in what way?

you have referenced papers, Belsky's for instance, that demonstrate that children with highly reactive (difficult) temperaments are more susceptible to parenting both for better and for worse. it's difficult to see why temperament (at least in terms of the reactive aspects of temperament) isn't measured until age 2, by which time the influence of the environment has already begun. I wonder if what is being measured from age 2 through 7 can really be considered temperament at this point or if it is rather the combination of the child's biological predispositions and its interaction with the rearing environment. This may be better characterized as the expression of the child's temperamental tendencies rather than temperament proper. or perhaps overall, you are meaning to discuss parents' perceptions of the children's temperament and no so much their actual behavior, in which case it should be referred to as perception of temperament.

lines 66 through 75, The research questions are not entirely clear. is the question whether the intervention group experiences changes in child temperament, changes in parents' perceptions of child temperament, or in parents' ability to successfully adapt to their children's temperament (thereby enhancing fit and subsequent child outcomes)? The general question in lines 76 through 78 does not follow conclusively from the preceding paragraph. It would be helpful if the research question/s were more specific, (for example, "associated" in what way?)

line 81, presumably the parents were participating in the parent training program, not the preterm infants :)

because you have indicated that a group of healthy full term infants was recruited, i think it would be meaningful to report how reported rates of "difficult" temperament compared between the preterm and full term groups just to orient readers who are not already familiar with how this may or may not look different in preterm infants.

Methods
It would be very helpful to have sample items, particularly for emotionality and soothability - I see there are examples in the discussion. I would move these to the methods section as that's likely where readers would go to look for them.
PSI - Abidin's PSI would have inherent in the measure of parenting stress, parents' perceptions of the child's difficultness. it might be important to make this clear.

line 200 - this sentence seems to state very clearly that you believe when changes were seen in reported child temperament, they reflect actual changes in children's temperament. is this what you mean to say? i find this to be controversial, especially in terms of the reactive (rather than regulatory) aspects of temperament, and therefore, think there should be more emphasis on this in the lit review. if instead, you mean to be arguing that the parenting intervention affected parents' self-efficacy or empowerment and therefore improved the way they receive and respond to their children's temperament, thereby altering their perception of their child's difficulty, then this first sentence of the discussion should perhaps be reworded to suggest that the study explores parents' perceptions of children's temperament over time. Alternatively, I might even eliminate the first sentence in the discussion and just start with the second one.

Overall, I feel the discussion goes back and forth between suggesting actual differences between groups or over time in children's temperament/behavior and that the parenting program improves parents' perceptions of their children's behavior. I'd like to know which the authors believe fundamentally and then have that be the consistent stance in the discussion as the framework from which to interpret and understand the findings. For me, it seems that the further explanations regarding how/why the intervention may have shown positive results suggest that this intervention impacted "goodness of fit" between parents perceptions and children's expressions of temperament. I find this to be a compelling argument and useful for practice and policy, but this is not an argument about altering children's temperament, but rather about an intervention to impact parents perceptions of children's "difficultness" (perhaps even how to effectively respond to and co-regulation children's reactive tendencies) as measured partly by their reports of temperamental expression and also inherent in their reports of parenting stress (as it's captured in Abidin's measure). If I have interpreted the material correctly, I think that could be made clearer in the first paragraph of the discussion.

In sum, I think this study is interesting and important and would make a worthwhile contribution to the field, particularly in terms of highlighting the importance of parents' perceptions of the children's behavior and also the ability to intervene and positively impact those perceptions. And I feel this could be made clearer to readers by being very specific in defining and naming the concepts being measured, analyzed, and discussed.

Author Response

Question: Thank you for the opportunity to review this article. The study presents meaningful questions and results relating to parents' perceptions of children's temperament, their associated parenting stress, and a parenting program designed to improve outcomes for parents and children. I appreciate the inclusion of both mothers and fathers as well as a parenting program with an intentional focus on temperament and parents' understanding of temperament. I would like to see some clarification of concepts as a means of making clear exactly what impact the intervention in purported to have. Primarily, this is a question of whether temperament in this study refers to children's actual behaviors or to parents' perceptions of those behaviors, so that when we read about change (or stability) over time, we understand whether you are suggesting the children are changing or the parents are changing.

Answer: This paper investigates mothers’ and fathers’ reports of children’s temperament on a standardized questionnaire at several ages. Thus, it reports parents’ perception of temperament and no observational data of temperament are available in this study. Observational data on child temperament could have been a useful supplement to this investigation. As mentioned by  Kopala-Sibley DC et at.al (2018), parents’ reports of child temperament are interesting because they draw on observations of the child over time and in different contexts, but they are found to be only modestly associated with observational measures of child temperament.

In line with this we consequently address the outcomes reported here as parents perception of child temperament, from the abstract to the conclusion.. After this review it is also added to the first sentence in the discussion  (line 200).

Question: i'd like to better understand the relationship between prematurity and what we traditionally think of as temperament. are there certain characteristics that are common to preterm infants that overlap with what we think of as temperament in full term infants? are these characteristics related to prematurity itself (some aspect less developed due to prematurity)? what i'm trying to ask is are we calling something temperament that is somehow different in preterm infants, related to a different biological underpinning than traditional conceptualizations of temperament in full term infants? This may not need to be explained in detail, but it seems like a question a reader might ask. Perhaps, this is just something to think about.

Answer: Very preterm born children are exposed to environmental experiences long before they are meant to. As a population they have been extensively reported in relation to temperamental dispositions. Despite this, we think that preterm born children are equipped with a similar biological fundament as any child. Early toxic stress may alter the development of their structural and functional brain development, and this may be related to reports of more internalized behavior for example. Even though, they develop temperamental expressions as a mix of biological and environmental conditions as any other child.

Question: lines 58 through 61, I don't understand how the preceding sentences lead to the conclusion in line 61. Also, what is the distinction between the child's actual temperament and parent's perception of the child's temperament. This seems very important to the argument. Are you arguing that the intervention impacts children's actual behavior (the manifestation of their actual temperament) or that it impacts parents' perceptions of the child's temperament? Clarification of this would be very helpful.

Answer: Several studies in this field of research discuss how relations might be between parenting stress, parents’ negative regard and parents’ perceptions of their child’s temperament. This might be especially important when a population of preterm born children is studied because these children are reported to be more dependent of a nurturing environment to achieve optimal development. We acknowledge that the reasoning might be difficult to follow and hope that it becomes easier if we include that this is especially related to preterm born children (preterm born is added in line 61).

Question: parenting stress - does this mean any sort of stress experienced by parents? or does this refer to Abidin's conception of parenting stress which is in large part a function of parents' perceptions of the difficulty of being a parent and parenting this specific child?

Answer: Abidin’s conceptualization of parenting stress is used in this study (presented in line 155-160). This measure captures both stress related to the child’s behavior, the parental role and the parent-child interactional patterns. Thus, we choose to use the total parenting stress measure as an independent variable in this analysis.

Question: line 65 - the program influenced the association between child regulation and maternal stress in what way?

Answer: As described in detail in the paper we refer to: Mothers, who had participated in the program and  cared for children characterized by high reactivity behavior reported higher levels of parenting stress than control mothers facing similar challenges at six month. At children’s age of one this relationship had changed, indicating a more successful adaptation between mothers and children in the intervention group than in the control group. To address this we have added “positively” to the sentence in line 66.

Question: you have referenced papers, Belsky's for instance, that demonstrate that children with highly reactive (difficult) temperaments are more susceptible to parenting both for better and for worse. it's difficult to see why temperament (at least in terms of the reactive aspects of temperament) isn't measured until age 2, by which time the influence of the environment has already begun. I wonder if what is being measured from age 2 through 7 can really be considered temperament at this point or if it is rather the combination of the child's biological predispositions and its interaction with the rearing environment. This may be better characterized as the expression of the child's temperamental tendencies rather than temperament proper. or perhaps overall, you are meaning to discuss parents' perceptions of the children's temperament and no so much their actual behavior, in which case it should be referred to as perception of temperament.

Answer: As mentioned above the paper uses the concept “parents’ perception of child temperament” across all chapters (after the inclusion in the first sentence of the discussion, 13 times).

Question: lines 66 through 75, The research questions are not entirely clear. is the question whether the intervention group experiences changes in child temperament, changes in parents' perceptions of child temperament, or in parents' ability to successfully adapt to their children's temperament (thereby enhancing fit and subsequent child outcomes)? The general question in lines 76 through 78 does not follow conclusively from the preceding paragraph. It would be helpful if the research question/s were more specific, (for example, "associated" in what way?)

Answer: To meet this need of clarification we have added the word “positively” to the research question. This seem appropriate as the direction of change/difference in parents’ perception of child temperament depends on the aspect assessed (e.g.  lowered emotionality seems preferable as all questions address aspects of negative emotionality, while higher degrees of soothability are associated with more shared pleasure in a family).

Question: line 81, presumably the parents were participating in the parent training program, not the preterm infants :)

Answer: In this program,  both parents and their child participated in all interventional sessions. Thus, it seems correct to write it this way even one or two parents participated.

Question: because you have indicated that a group of healthy full term infants was recruited, i think it would be meaningful to report how reported rates of "difficult" temperament compared between the preterm and full term groups just to orient readers who are not already familiar with how this may or may not look different in preterm infants.

Answer: The author’s discussed if we should include data on termborn control children in this study but decided  not to do that. 

Question: Methods 
It would be very helpful to have sample items, particularly for emotionality and soothability - I see there are examples in the discussion. I would move these to the methods section as that's likely where readers would go to look for them. 

Answer: Descriptions related to the emotionality and soothability dimensions of children’s temperament are moved from the discussion section to the description of measures, line 140-144.

Question: PSI - Abidin's PSI would have inherent in the measure of parenting stress, parents' perceptions of the child's difficultness. it might be important to make this clear.

line 200 - this sentence seems to state very clearly that you believe when changes were seen in reported child temperament, they reflect actual changes in children's temperament. is this what you mean to say? i find this to be controversial, especially in terms of the reactive (rather than regulatory) aspects of temperament, and therefore, think there should be more emphasis on this in the lit review. if instead, you mean to be arguing that the parenting intervention affected parents' self-efficacy or empowerment and therefore improved the way they receive and respond to their children's temperament, thereby altering their perception of their child's difficulty, then this first sentence of the discussion should perhaps be reworded to suggest that the study explores parents' perceptions of children's temperament over time. Alternatively, I might even eliminate the first sentence in the discussion and just start with the second one.

Answer: I hope it has become clear in the previous comments that this paper report findings related to parents perception of children’s temperament. To what degree differences and trends across different ages reflect changes in characteristics of children’s behavior styles or the parents perceptions is not investigated as there does not exist (as far as we know) a gold standard to measure temperament. What seems reasonable is that  differences in perceptions may reflect different levels of adaptation and healthy functioning in families, even this is not the scope of this paper. As described in the introduction we think that the transactional nature of adaptations between parents and their child is important to keep in mind when reflecting upon these theoretically based conceptualizations of child behavior.

As mentioned above, we have clarified the first sentence of the discussion.

In sum, I think this study is interesting and important and would make a worthwhile contribution to the field, particularly in terms of highlighting the importance of parents' perceptions of the children's behavior and also the ability to intervene and positively impact those perceptions. And I feel this could be made clearer to readers by being very specific in defining and naming the concepts being measured, analyzed, and discussed.

Reviewer 2 Report

This was a comprehensive study with excellent longitudinal support and the finding that soothability and emotionality were different but sociality, shyness and activity were not adds to the strength of the finding on soothability sociality were effected by the intervention.

The presentation of the data, specifically Tables 4 and 5, should be clarified.

One concern re the statistics; the authors report:

"Separate analyses were conducted for each dimension of temperament and for reports from mothers and fathers, thus ten analyses were performed. Time was treated as a continuous variable, coded as the number of years from baseline, and baseline was defined as the measure of temperament at age two (time coded as 0, 1, 3, 5).  Intercepts were allowed to vary between subjects, and a random slope for time was included in the model. The level of significance was 0.05."

Did the authors correct for multiple analyses?

Author Response

Question: This was a comprehensive study with excellent longitudinal support and the finding that soothability and emotionality were different but sociality, shyness and activity were not adds to the strength of the finding on soothability sociality were effected by the intervention.

The presentation of the data, specifically Tables 4 and 5, should be clarified.

Answer: Thank you for your comments. We chose to include the tables 4 and 5 (as additional information online) because they display the impact of covariates included in the analyses. Table 3 describes the significance of the time and group variables for each dependent variable (the five dimensions of temperament reported from mothers and fathers separately). The headings of table 4 and 5 are modified to describe this.

Question: One concern re the statistics; the authors report: "Separate analyses were conducted for each dimension of temperament and for reports from mothers and fathers, thus ten analyses were performed. Time was treated as a continuous variable, coded as the number of years from baseline, and baseline was defined as the measure of temperament at age two (time coded as 0, 1, 3, 5).  Intercepts were allowed to vary between subjects, and a random slope for time was included in the model. The level of significance was 0.05." Did the authors correct for multiple analyses?

Answer: As described in the result section and in the tables we have done 5 separate analyses on data reported from mothers and  5 separate analyses on data reported by fathers and we have not corrected for multiple analyses. It seems necessary to analyze reports from mothers and fathers separately and in addition, the analyses cover five separate dimensions of child temperament.   To clarify this we have added some sentences at the ending of the discussion.

References in this reply to the reviewers.

Kopala-Sibley DC, Olino T, Durbin E, Dyson MW, Klein DN. The stability of temperament from early childhood to early adolescence: A multi-method, multi-informant examination. Eur J Pers. 2018;32(2):128-45.